# Landslide Susceptibility Assessment Considering Landslide Volume: A Case Study of Yangou Watershed on the Loess Plateau (China)

**Hang Gao [1] and Xia Zhang [2],***

[1]  Faculty of Infrastructure Engineering, Dalian University of Technology, Dalian 116024, China; hgao_muriel@foxmail.com
[2]  State Key Laboratory of Eco-Hydraulics in Northwest Arid Region, Xi'an University of Technology, Xi'an 710054, China
*   Correspondence: zhangxia20002000@163.com

**Abstract:** Because of the special geological conditions on the Loess Plateau, Landslide erosion is not only the main goal of prevention and control of geological disasters, but also an important erosion mode of soil and water loss in the basin. Thus, landslide susceptibility assessment before only considering landslide frequency is not far enough for a decision-maker. The study aims to consider both frequency and scale of landslides for a better landslide susceptibility evaluation. Taking the Yangou small watershed as an example, this study used a VR model, RIRA method, and the GIS method to comprehensively consider frequency and scale to analyze landslide susceptibility of the small watershed. Based on the detailed analysis of the existing literature, slope, elevation, NDVI, land-use, lithology, amount distant to road, amount distant to river, profile curvature, and rainfall as landslide are selected as the conditioning factors (CFs) of the landslide, to draw the sensitivity map. The map of landslide susceptibility was classified into five zones: very low, low, medium, high, and very high, and the cover areas occupy 6.90, 12.81, 12.83, 9.42, and 5.87 km$^2$, respectively. A total of 60% of the landslide occurred in the zones of high and very high susceptibility, accounting for 87% of the total volume in the study area. The very high susceptibility is the area with a larger relief and along the river and road. The findings will help decision makers to formulate scientific comprehensive policies that take into account disaster prevention and soil conservation measures in specific regions.

**Keywords:** landslide susceptibility; volume ratio; reserve increase-rate-analysis method; GIS; Loess Plateau

## 1. Introduction

Landslides, a mass failure on steep slopes, facilitated by gravity are an important process controlling the sedimentary structures and growth patterns of the steep slope [1], and major natural disasters that often cause human and economic losses due to natural forces and human actions [2]. There are mainly three manifestations such as falling, sliding, or flowing [3]. China is covered by a vast area that is considered to be one of the most landslide-prone regions in the world, while landslide phenomena cause an estimated 700 to 1000 deaths every year and more than RMB 10 billion annually in infrastructure and property damage [4]. Landslides, especially on the Loess Plateau, have been widespread due to their topography and geographical location. In order to prevent and mitigate the damage, it is necessary to need comprehensive information regarding landslides, such as occurrence, spatial distribution, and susceptibility. Landslides also deliver huge amounts of sediment to rivers in the mountainous and hilly watersheds [5,6]. In particular, most landslides occur on slopes, whether man-made or natural, where loose material is always transported to the toe and downhill, which may have an out-of-field effect on sediment

transport through river flows to the downhill [7]. On the Loess Plateau, the phenomenon is particularly serious, which is one of the reasons for the poor soil erosion in the region due to unmanageable, heavily containing sediment flow [8]. It can be seen that landslides have a great effect on water and soil loss in the basin. Older land use policies may not always reflect optimal planning for land use prone to landslides [9]. A reasonable zoning prediction of regional landslides can therefore assist planners and engineers in making decisions on the use of such lands [6].

Landslide susceptibility is recognized as the propensity of an area to generate landslides [10]. Generally, landslide occurrence in each region is a function of various factors. Each of the factors and function has a different influence. In order to assess susceptibility from gravity, it is, therefore, necessary to identify and analyze the factors leading to gravity [11]. The occurrence of landslides is controlled both by a series of predisposing factors, e.g., geological, geomorphological, climatic and hydro-geological, and triggering factors, including seismicity, heavy rainfall, human activities, and freeze-thaw [12]. Although it is still difficult to predict a landslide event in space and time, a region may be divided into sub-regions with homogenous properties, which are classified and ranked according to the potential hazard degrees of group movement considering the prerequisite factors explained above. A landslide susceptibility map is considered as an effective solution, and it is also an important task for decision making on regional planning and protection.

A variety of methods can be employed to develop the landslide susceptibility assessment. There are two main types: a "knowledge-driven model" that uses relevant domain knowledge for analysis and a "data-driven model" that uses intelligent methods to automatically find patterns from data. The former methods include fuzzy logic [13], fuzzy comprehensive evaluation [14], and the analytic hierarchy process [15]. The latter methods include logistic regression analysis [16], decision tree [17], random forest [18], artificial neural network [19], and support vector machine [20]. In general, the knowledge-driven model is more subjective while the data-driven model is relatively objective and can accurately reflect the correlation between landslide susceptibility and its basic environmental factors. However, the data-driven model is complex in the model training and testing process and requires a large number of training samples. Both types of models can get a more reasonable result, however; they mainly provide information on the probability of landslide occurrence, but no information on the size of a potential landslide once occurring. In other words, the landslide susceptibility map applied by those methods only reflected the possibility of landslide occurrence, while it cannot express the probability of a landslide. Research has shown that there is a significant difference in spatial variation between the susceptibility and erosion risk of a mass movement in the watershed [21]. Thus, the results only consider landslide frequency as incomplete for the decision-makers to implement land planning or other policy in the region. Therefore, it will also be meaningful to understand the sediment yield and prevent soil erosion in the catchment when both considering the scale and the frequency of the landslide.

If the volume of the landslide is considered in the assessment of landslide susceptibility, it is necessary to choose a suitable method. Whatever the relationship between landslide and the landslide-related factors, it can be inferred from the relationship between landslide and the landslide-related factors that had occurred. Thus, volume ratio (VR), which reflected the relationship between the volume of the landslide and each landslide conditioning factors, is used in the study, and the revised increase-rate-analysis method (RIRA) method is also a suitable way for quantitative analysis of the relationship between landslide scale and various factors. The RIRA method is an improvement on the increase-rate-analysis method (IRA), which is used to evaluate the sensitivity of the landslide to a single variable [1]. On this basis, Lucas [22] has proposed a revised increase-rate-analysis method (RIRA) based on the importance ranking of influencing factors and sensitivity analysis for a large number of data-influencing factors. The method has been successfully applied in the analysis of the influencing factors of the silt dam conditions in the Culiacan basin in northwestern Mexico. Recently, as a basic analysis tool for landslide hazards,

ArcGIS can effectively perform spatial data management and manipulation for the analysis and has been used in many studies [16]. It is also an indispensable tool for landslide susceptibility mapping.

The objective of this study included: (a) applying the VR model, the RIRA method, and GIS in the spatial event prediction based on considering landslide scale; (b) creating a landslide susceptibility map for the Yangou small watershed on the Loess Plateau of China; (c) assessing the landslide susceptibility and providing suggestions for landslide prevention and soil erosion of the small catchment.

## 2. Study Area

The study focuses on the Yangou watershed, a small catchment located in Yanan city in the north Shanxi province of China (Figure 1). It is enclosed within the latitudes 36°28′ N and 36°34′ N and longitudes 109°27′ E and 109°33′ E, covering an area of about 48 km². The catchment is typical of the gullied-hill zone and is located in the central area of the Loess Plateau, which is a sub-watershed in the Yanhe basin. The river is a secondary branch of the Yan River and flows from southeast to northwest. The watershed exhibits complex topographic variations, with a gully density of 4.8 km km$^{-2}$ and an elevation ranging from 988 m to 1404 m. The terrain gradient of the basin is mostly composed of steep hill slopes. The slope analysis showed that the slope distribution in the study area ranged from 0° to 51°. The climate belongs to the transitional zone from warm semi-dry to semi-wet. The average annual precipitation is 550 mm, and 70% of the annual precipitation occurs from June to September. Generally, the southern section had higher values than the north. The geotechnical medium in the area can be divided into rock mass and soil mass, mainly sandstone and loess. The saturated bulk density (γ), Poisson's ratio, cohesion (C), and friction angle (φ) of sandstone are 2.68 g/cm³, 0.22, 4.23 MPa, and 41.6°, respectively. The moisture content (ω), dry bulk density (γ), cohesion (C), and friction angle (φ) of loess are 15.6%, 1.51 g/cm³, 39.48 MPa, and 26.95°.

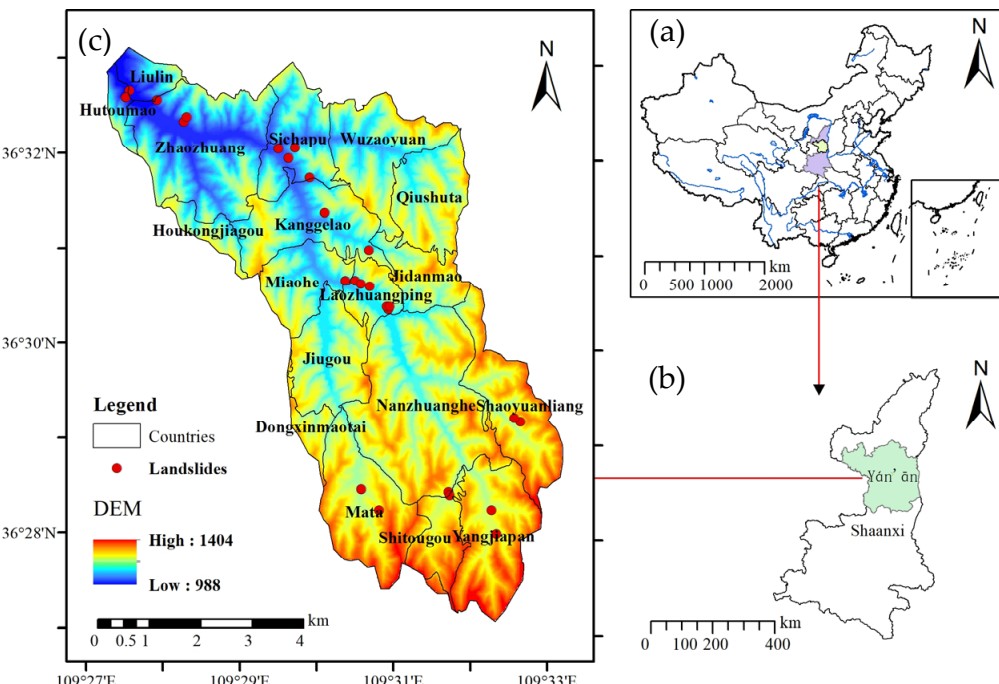

**Figure 1.** Study area in China. (**a**) Shaanxi province in China; (**b**) Yan'an city in the Shaanxi province; (**c**) Yangou watershed in the Yan'an city.

## 3. Methods

The landslide susceptibility maps were prepared using VR and RIRA models to assess the landslide susceptibility in the Yangou catchment. Figure 2 showed the flow chart of the application method, including the data collection, gathering of the landslide conditioning factors, determination of relationship between each condition factors' subclasses, and landslide inventory map using VR, application of the RIRA method in the landslide sensibility, landslide susceptibility assessment of numerical data integration, and landslide susceptibility mapping.

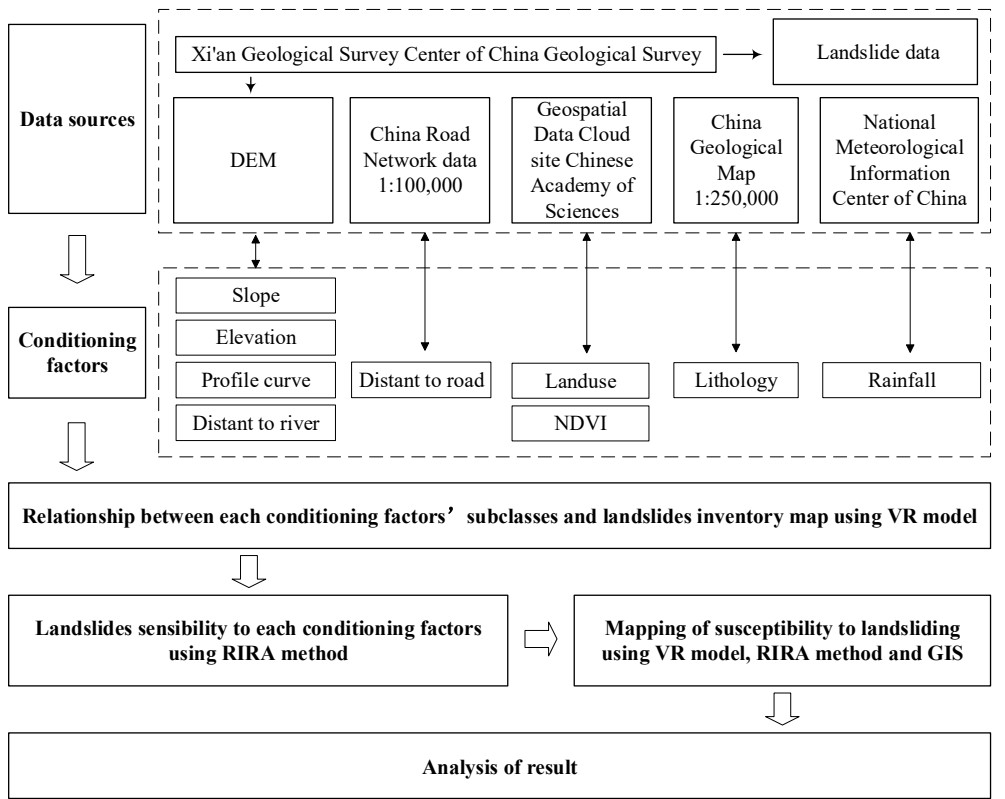

**Figure 2.** Flowchart of the used methodology.

The landslide inventory database, which included 27 locations, volumes, and types were provided by the Xi'an Geological Survey Center of the China Geological Survey. The landslide is divided into four sections: $\leq 1 \times 10^4$, $1 \times 10^4$–$10 \times 10^4$, $10 \times 10^4$–$100 \times 10^4$, and $\geq 100 \times 10^4$ m$^3$, corresponding to tiny, small, medium, and large landslides. The amount of the landslide and its percentage in each interval were counted. In this study, nine parameters were chosen according to the literature review and the general environment of the study area. These conditioning factors are land-use, slope, profile curvature, altitude/elevation, distance to roads, distance to rivers, lithology, normalized difference vegetation index (NDVI), and rainfall. The altitude/elevation, slope, and profile curvature were extracted from the Digital Elevation Model (DEM) (cell size: 25 m × 25 m), which is converted from a topographic map with a scale of 1:100,000. They were also collected from the "Xi'an Geological Survey Center of China Geological Survey". The NDVI was extracted from the Landsat 8/OLI data set, which is provided by the Geospatial Data Cloud site, Computer Network Information Center [23]. The computational formula of NDVI is

$$\text{NDVI} = (NIR - R)/(NIR + R) \tag{1}$$

where *NIR* is the reflection value of the near-infrared band, and *R* is the reflection value of the red band. The landuse data were provided by the Resource and Environment Science and Data Center [24]. It was decided that the study area was mainly covered by eight

land-use types, namely water, wetland, farmland, resident land, bareland, forest, shrubland, and grassland. The lithology of the Yangou Watershed was delineated using the China Geological Map at a 1:250,000-scale. The distance to rivers was extracted using hydrological analysis of DEM and neighborhood statistic in the ArcGIS. The distance to roads was developed using China Road Network data at a 1:100,000-scale. The rainfall data were from the China meteorological background data set of a 500 m × 500 m pixel size, which is provided by the Resource and Environment Data Cloud Platform [25]. Then, these nine conditioning factors were selected in the present study and were standardized to the same size of 25 m × 25 m for further analyses. The study area covers 355 columns and 447'rows. The spatial analysis tool in the ArcGIS software was used to count the corresponding values of each factor at the landslide location. Also, the occurrence frequency ratio of the different classes was calculated in the land use and geology as the corresponding values of the landslide.

To show the relationship between each factor's subclasses and landslide volumes, the VR approach is proposed. It is a variant of the probabilistic method that is based on the observed relationships between the volume of the landslide and each landslide conditioning factor. The volume ratios for the classes or types of each conditioning factor were calculated by dividing the landslide volume by the typical value ratio of the study area. Each factor's ratio value was calculated using Equation (2):

$$VR_{jk} = \frac{V_{jk}^*}{V_{jk}} / \frac{T^*}{T}$$ (2)

$VR_{jk}$ = volume ratio of each factor; $V_{jk}^*$ = volume of observed landslide of class "*k*" of factor "*j*"; $V_{jk}$ = volume of class "*k*" of factor "*j*"; $T$ = typical value of the map, such as the total area of the map, the average distance to roads or rivers in each class, the average elevation of each class, and the average rainfall of each class. $T^*$ = typical value of the observed landslide of the map, such as the total area of each class, the average distance of the landslide to roads or rivers, the average elevation of the landslide, and the average rainfall of the landslide. A value of 1 is the neutral value, and the values higher than 1 show a high positive correlation with a certain factor of the landslide [26]. A higher VR value shows that a higher probability of the landslide volume occurs.

The RIRA is a method of sensitivity analysis used by calculating the relationship between dependent variables and independent variables. RIRA can assess the relative contribution of each variable by placing greater emphasis on known variables contributing to landslides, and which are sorted by contribution [22]. In addition, each variable has a specific value in the method, which ensures the objectivity of the data calculation. To eliminate the dimensional impact between the independent variables, all data of the factors were standardized using the z-scores method before calculation. The dependent variables (such as the amount of landslide) and the independent variables (such as profile curvature, slope, altitude, NDVI, land-use, geology, distance to rivers, distance to roads, and rainfall) were listed separately. The variables were sorted in order from small to large, and the sensitivity coefficients of each influencing factor were calculated. In order to quantify the impact of landslide-related factors on the landslide, the calculation steps of RIRA were as follows: first, the landslide increase rate (%) was calculated using Equation (3):

$$RP_i = \frac{2(P_i - P_{i-1})}{P_i + P_{i-1}}$$ (3)

where $P_i$ is the amount of the *i*-th landslide (10⁴ m³). Then, the increase rate (%) of each impact factor on the landslide was calculated as follows:

$$Rt_i = \frac{2(t_i - t_{i-1})}{t_i + t_{i-1}}$$ (4)

where $t_i$ and $t_{i-1}$ are the values of the explanatory variable that are sequentially connected. $Rt_i$ is the increase rate calculated for two successively independent variables ordered by the values. Therefore, the Absolute Sensitivity Parameter (s), towards an independent variable, $t$, was calculated by its mean growth rate:

$$s_{aj} = \left| \overline{(RP_{i,j}/Rt_{i,j})}_{i=1,N} \right| \tag{5}$$

Being $j$ the $j$-th independent variable, $N$ the total number of landslide data, namely slope, aspect, altitude, NDVI, land use, geology, distance to rivers and distance to roads. A factor with a larger absolute value of the s represents the higher sensitivity of the amount of the landslide to the related factor. In the study, the revised increase-rate-analysis (RIRA) method is used to assess the relationship between the factors and landslide.

At last, the landslide susceptibility of each subclass was calculated by Equation (6):

$$S_{ak} = s_{aj} \times VR_{jk} \tag{6}$$

$S_{ak}$ is the susceptibility of each $k$-*th* class. Then, each class susceptibility map can be obtained using the $S_{ak}$ value in the ArcGIS software. At last, all thematic layers were combined using the $S_{ak}$ value to derive the landslide susceptibility map of the Yangou watershed. Calculated susceptibility values were classified into areas of very low, low, medium, high, and very high susceptibility using the natural break method available within the ArcGIS software.

## 4. Results and Discussion

### 4.1. Landslide Inventory

Landslides in the study area primarily lie along the northwest and southeast direction. There are two types of landslides in the region: slides and falls. About 15 observed landslides belong to slides and 12 falls. The amount of erosion is $704 \times 10^4$ m$^3$ and $14.4 \times 10^4$ m$^3$, accounting for 98% and 2% of the total landslide, respectively. The slide is the main type of landslide in the region, which had an important contribution to the frequency of landslide and the total amount of slope movements in the region. The triggering mechanism of falls is different from that of slides. When the water in the gully scours and transports the lower accumulation, and the gravity moment generated by the upper steep wall under the action of gravity is greater than the tensile force moment of the soil, the upper soil layer loses its balance to fall [27]. There were many small falls in the study area. Most of the fall-bodies fell and broke, which were difficult to preserve for a long time. Most of them have been transported to form sand production, which is difficult to investigate [28].

The volume of the smallest landslide is approximately $0.2 \times 10^4$ m$^3$, the largest is around $270 \times 10^4$ m$^3$, and the average is estimated to be $26.6 \times 10^4$ m$^3$. Figure 3 showed the amount and frequency of landslides in each interval, which were tiny, small, medium, and large landslides. The total amount of erosion caused by large landslides of $\geq 100 \times 10^4$ m$^3$ was the largest, which was $503.6 \times 10^4$ m$^3$, accounting for 70.2% of the total erosion and followed by the erosion of the medium landslide, which was $174.7 \times 10^4$ m$^3$, accounting for 24.4% of total erosion. The small landslide was $37.1 \times 10^4$ m$^3$, accounting for 5.2% of total erosion, and the tiny landslide had the least landslide, at $2.7 \times 10^4$ m$^3$, accounting for 0.2% of total erosion. In the case of frequency of landslides, there were 19 small-scale erosions with a volume of fewer than $10 \times 10^4$ m$^3$ in the study area, accounting for 70% of the total frequency of erosion, that is, the occurrence of tiny-scale and small-scale landslides were higher in the area. While the amount of medium-sized and large-scale collapse accounted for 94.6% of the total, their frequencies of occurrence were only 5 and 3 times, accounting for 19% and 11% of the total frequency, respectively. It can be seen that the number of tiny–small landslides in the study area was more frequent. The frequency of medium and large-scale landslides was little, but they had a major contribution to the total amount of landslides. It can also be seen from the statistics that the contribution of frequency

and the volume of landslides are not the same, and it is more meaningful to consider both effects in the study. The cumulative amount of landslide erosion in the study area reached $718.1 \times 10^4$ m$^3$. A large amount of landslide erosion has an important impact on sediment yield and sediment transport in the basin. The landslide-triggered erosion enters the channel and then flows into the river system by runoff, directly or indirectly transporting a large amount of sediment to the channel, which is one of the main processes of soil erosion and river sediment sources. Large slides also cause road damage and soil erosion. The volume of the falls is not large but their migration speed is fast. Because the occurrence to stop movement does not exceed 1 min, people are often unable to hide and suffer the risk of casualties; the harm is no less than the landslide. Between 1980 and 2015, there were 53 landslides over the Loess Plateau, killing and missing 717 people [29].

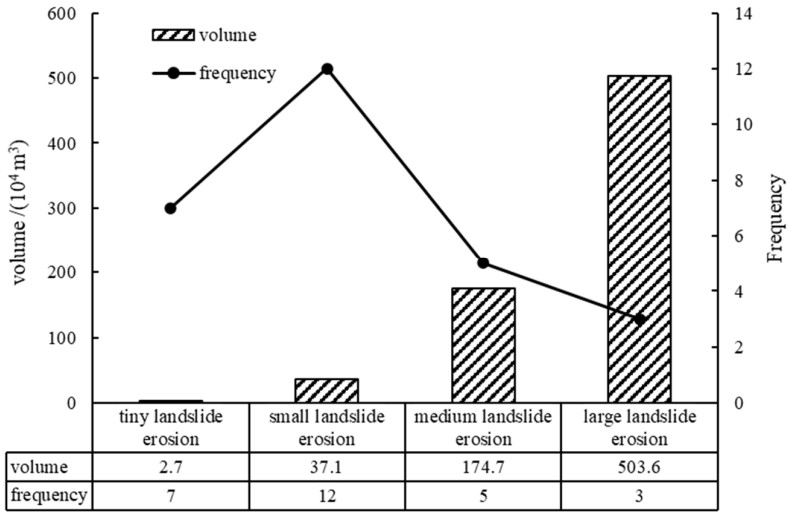

**Figure 3.** Amount and frequency of landslides in the Yangou watershed.

### 4.2. Application of the VR Model

In order to study, in detail, the relationship between each factor and the scale of the landslide, each factor is divided into several classes. The results of the spatial relationship between landslide volumes and the classes of landslide conditioning factors using the VR model are shown in Table 1.

**Table 1.** Spatial relationship between landslide conditioning factors and landslide by volume ratio model.

| Independent Variable | Classes | $V_{jk}^*/V_{jk}$ | $T^*/T$ | $VR_{jk}$ |
|---|---|---|---|---|
| Slope angle (°) | 0–15 | 0.7294 | 0.8629 | 0.8453 |
| | 15–25 | 0.2427 | 0.1272 | 1.9089 |
| | 25–35 | 0.0259 | 0.0094 | 2.7538 |
| | >35 | 0.0019 | 0.0006 | 3.5264 |
| Elevation (m) | <1000 | 0.1448 | 0.9991 | 0.1450 |
| | 1000–1100 | 0.0947 | 0.9874 | 0.0959 |
| | 1100–1200 | 0.1886 | 0.9946 | 0.1896 |
| | 1200–1300 | 0.5719 | 0.9937 | 0.5755 |
| | >1300 | 0.0000 | 0.0000 | 0.0000 |
| NDVI | no cover | 0.0000 | 0.0004 | 0.0000 |
| | very low cover | 0.1478 | 0.0263 | 0.1778 |
| | low cover | 0.7124 | 0.2459 | 0.3451 |
| | medium cover | 0.1398 | 0.5355 | 3.8301 |
| | high cover | 0.0000 | 0.1919 | 0.0000 |

**Table 1.** *Cont.*

| Independent Variable | Classes | $V_{jk}^{*}/V_{jk}$ | $T^{*}/T$ | $VR_{jk}$ |
|---|---|---|---|---|
| Land-use | Farmland | 0.2981 | 0.3063 | 0.9734 |
| | Forest | 0.0199 | 0.4000 | 0.0498 |
| | Grassland | 0.4029 | 0.2345 | 1.7180 |
| | Shrubland | 0.0000 | 0.0021 | 0.0000 |
| | Wetland | 0.0000 | 0.0000 | 0.0000 |
| | Water | 0.0000 | 0.0006 | 0.0000 |
| | Resident land | 0.1313 | 0.0367 | 3.5780 |
| | Bareland | 0.1478 | 0.0198 | 7.4512 |
| Lithology | Sandstone | 0.4169 | 0.2848 | 1.4640 |
| | Loess | 0.5831 | 0.7152 | 0.8152 |
| Distant to road (m) | <50 | 0.5289 | 0.4101 | 1.2897 |
| | 50–150 | 0.2454 | 0.9813 | 0.2501 |
| | 150–300 | 0.0100 | 0.8237 | 0.0122 |
| | 300–500 | 0.1824 | 0.9151 | 0.1994 |
| | 500–800 | 0.0333 | 1.1883 | 0.0280 |
| | >800 | 0.0000 | 0.0000 | 0.0000 |
| Distant to river (m) | <50 | 0.7368 | 1.1093 | 0.6642 |
| | 50–150 | 0.2543 | 0.9612 | 0.2645 |
| | 150–300 | 0.0089 | 0.8106 | 0.0110 |
| | >300 | 0.0000 | 0.0000 | 0.0000 |
| Profile curvature | Concave slope | 0.5151 | 0.5384 | 0.9567 |
| | Convex slope | 0.4849 | 0.4616 | 1.0505 |
| Rainfall | Less rainfall area | 0.2051 | 0.9998 | 0.2052 |
| | Medium rainfall area | 0.3779 | 1.0002 | 0.3779 |
| | More rainfall area | 0.2415 | 0.9996 | 0.2416 |
| | Heavy rainfall area | 0.1755 | 0.9996 | 0.1755 |

In the case of slope angle, the results showed that the VR values also increased as the slope increased, indicating that the slope angle has a good positive correlation with the probability of the landslide volume. The slope is a major factor in landslide occurrence because it relates to drainage, fault line, and road networks [2]. In the Yangou watershed, the 0°–15° slope area constituted 20.34% of the total area, the 15°–25° area accounted for 35.49%, the 25°–35° area accounted for 36.57%, and slope areas of more than 35° were 7.61% of the total area. The findings revealed a wide range of slope changes across the whole research area. According to the survey, 78% of the landslides occurred on slopes greater than 15° in the study area. Generally, the mass movement shifts from sliding to slumping with an increasing angle [30,31]. Thus, landslides are created in a hilly region or mountainous areas with steep slopes. The slope affects the scale and intensity of surface material flow and energy conversion. For example, the slope affects the area and amount of rainfall on the slope, thereby affecting the magnitude of runoff, infiltration, and runoff kinetic energy [32]. Thus, the greater the slope, the greater the gravity erosion [20,33].

There is no correlation between landslide volume and elevation rise in terms of elevation. There are many reasons for the influence of elevation on the development of landslide disasters, such as that different elevation ranges have different climatic characteristics and local depressions are due to slope differences, whether there is a sliding surface and an intensity of human activities in different elevation ranges [34]. Thus, elevation has no direct impact on the distribution of gravity erosion disaster points in the region. The 1200–1300 m area had the highest VR value of 0.5755, followed by 1100–1200 m (0.1896), and <1000 m (0.1450). The 1000–1100 m area has the lowest value of 0.959. In the Yangou watershed, 96.3% of the landslides occurred on the altitude ranging from 1000 m to 1300 m. This showed that landslides usually occurred at intermediate elevations. The results can also be

found in the other places since slopes tend be covered by a layer of thin colluvium that is prone to landslides [35,36].

The NDVI is a measure of surface reflectance to give a quantitative estimate of biomass and vegetation growth [37]. The value of NDVI in the study area ranges from −0.27 to 0.59 in the summer. Generally, the larger the NDVI value is, the higher the vegetation coverage is. In the case of NDVI, there is no landslide in the classes of no cover and high cover. In the other classes, the VR value reflected the probability of landslide volume increased with the increase of the plant cover. In other words, the vegetation didn't play important roles in the decrease of the landslide amount. There are positive and negative effects of vegetation cover on landslides. Vegetation roots can reduce the creep of the soil layer by increasing the shear strength of soil, which is negatively correlated with the amount of landslide erosion. On the other hand, in the field investigation of small watersheds, it was found that with the increase of vegetation coverage, the runoff on the slope increased, thereby increasing the gully erosion of the watershed [38]. Plant root splitting may also promote the occurrence of landslide erosion.

Land-use is another factor used to consider the natural and man-made environmental impacts on the land surface. The main land use types in the study area were forestland and cropland, accounting for 40% and 31% of the total study area, respectively. Secondly, the grassland accounted for 23% of the total study area. Other classes accounted for a combined 6% of the total study area. The bareland had the highest VR value of 7.4512, followed by resident land (3.5780) and grassland (1.7180). The class of forest and farmland had the VR values of 0.9734 and 0.0498, respectively. In the shrubland, wetland, and water, the VR values are zero. The results indicated surface attachment plays an important role in slope stability compared with bareland, and in our case, the cultivated areas are less prone to landslides when compared to other land cover types. This may be due to positive impacts of land management with better water and soil control measures on slope stability [39].

Different geological formations have different compositions and structure, which contribute to the strength of the material. Stronger rocks are more resistant to the driving force of landslides and less prone to damage than weaker rocks [40]. The main lithology units of the study area consisted of Jurassic sandstone and Quaternary loess. Vertical joints, structural joints, collapse joints, and weathering joints of loess are generally developed, which are potential geological factors leading to geological disasters such as loess slides and falls. The deformation and failure caused by loess collapsibility provide a channel for precipitation collection and rapid infiltration, and often lead to geological disasters such as landslides. The vertical joints and weathered joints of sandstone in the slope zone are developed, and the development and expansion of joint cracks form the dangerous rock mass of slope is seen, which often leads to fall disasters [25]. The joints in all the study area are very developed. From the VR values, it is seen that the high landslide volume occurred in the lithological units of sandstone with the VR value of 1.4640, followed by the lithological unit of loess with the VR value of 0.8152. The Jurassic sandstone had a higher value because it is mainly in the valley area; as the drainage area, the hydrological conditions are more complicated.

The stability of the soil mass might be harmed by road cutting caused by slope excavations. As a result, one of the influencing elements for landslides is closeness to a road. In the case of distance to roads, distance to roads of 0–50 m had the highest VR values, while the VR value in the distance to roads of 300–500 m had saltation. The result may be that the population density is high in the study area [41], and thus the road net is also density. In the range of 300–500 m, large collapses had occurred. However, the overall trend is that the VR value decreased as the distance increased, indicating that the distance to roads has a high influence on landslide occurrence and scale. Actually, the amount of landslide occurrence within 150 m of the road accounted for more than 78% in the study area.

There are reverse correlations between landslide volume and parameters based on the VR value of the distance to river. In the study area, 89% of the landslide occurred

within 150 m of streams. At the same time, the hydrographic axes of the third and fourth order streams are considered to be the main factor of the occurrence of landslides [28]. This is because the scouring effect of river flow will erode the slope. At the same time, the seepage effect of water flow makes the slope soil saturated. A large amount of free water is accumulated in collapsible loess, and the shear strength of soil is reduced, which has a negative impact on the slope stability. In addition, human activities are mostly distributed in the valley, which reduces the slope stability and is also the cause of frequent disasters and serious landslide erosion on both sides of the river.

Profile curvature measures the rate of change of slope, which controls erosion and deposition by affecting the acceleration and deceleration of the flow across the surface. Convex surface is positive and concave surface is negative [42]. For the profile curvature, 44% of the landslide occurred on the concave slopes, while 56% occurred on the convex slopes, and the VR value for convex slope (1.0505) is greater than that of the concave slope (0.9567), indicating that the convex slope has a little more effect on the volume of landslides. The concave slope easily collects rainwater, and rainwater infiltrates along the slope, which reduces the shear strength of soil and causes the instability and deformation of soil. At the same time, the potential energy of convex slope soil is closer to the limit value of mechanical equilibrium, and is also vulnerable to stronger weathering, which may also lead to the occurrence of slope instability. Stochastic statistics of 300 unstable slopes in China have shown that the instability of convex slopes is better than that of concave slopes [43]. Besides, in the experiment, it is verified the larger volume of landslides easily occurred in the convex slope [44]. The failure characteristics of loess soil slope are creep, and soil creep tends to produce convex terrain [45]. Therefore, the convex slope is dominant in the steep loess slope, and it is also prone to landslides [46].

The precipitation intensity is an important factor to affect the occurrence of landslides in the semi-arid region, although the interval between minimum and maximum rainfall is small [40]. In the case of rainfall, the medium rainfall area had the highest VR value of 0.3799, followed by more rainfall area (0.2416), less rainfall area (0.2052), and heavy rainfall area (0.1755), which indicated that there is no cear relationship between the amount of this factor and landslide volume. On the Loess Plateau, long-term and strong early effective rainfall both triggered clusters of shallow landslides in the previous study [47].

### 4.3. Application of the RIRA Method

The sensitivity coefficient of different related factors on landslides is quite different. The results of the sensitivity of the slope, elevation, NDVI, land-use, lithology, distance to road, distance to river, profile curvature, and rainfall on the volume of landslides performed by RIRA are reported in Table 2. The larger the sensitivity of an independent variable, the more prominently this variable influences the volumes of landslide. Profile curvature was found to be the most important influential factor, as it is the most sensitive parameter (0.3616), followed by the distance to river (0.2567) and distance to road (0.0908). Rainfall was the fourth in the ranking of the relative sensitivity, and the value is 0.0628. The relative sensitivity of elevation, slope, and NDVI are comparable to each other, although these variables are less important than the three variables motioned above (0.0523, 0.0345, and 0.0203, respectively). Finally, lithology and land use have relatively little relative sensitivity.

**Table 2.** Sensitivity analysis of landslides and related factors in the Yangou watershed (China).

| Parameter | Independent Variables | | | | | | | | |
|---|---|---|---|---|---|---|---|---|---|
| | Profile Curvature | Distant to River | Distant to Road | Rainfall | Elevation | Slope | NDVI | Lithology | Land-Use |
| Absolute Sensitivity | 0.3616 | 0.2567 | 0.0908 | 0.0628 | 0.0523 | 0.0345 | 0.0203 | 0.0099 | 0.0097 |

In summary, the independent variables, profile curvature, distance to river, distance to road, rainfall, elevation, and slope, have an important impact on the triggering and scale of landslides. Especially, both the relative sensitivity of two variables, profile curvature, and the distance to river, are very high. Convex and concave slope profiles are common in natural hill slopes. In the Yangou watershed, it is the most import factor. That means

landslides in areas with large undulating terrain are subject to key monitoring. Therefore, the influence of slope morphology should be taken into account when humans are carrying out activities. In the case of the relationship between landslides and distance to streams, the VR value rises as the distance to a stream decreases. The river has a strong erosive effect on the exposed area of the bedrock, which affects the stability of the valley slope. Especially in smaller valleys, both the down erosion and side erosion of flowing water exist, and the valley slopes on both sides are steep, which are in the erosion of flowing water and are high-risk areas of slides and falls. In addition, the most direct impact of long-term surface water accumulation is the rise of the groundwater level, which forms groundwater in a large range and affects slope stability through groundwater. The groundwater on both sides of the basin passes can aggravate the slope instability by softening and eroding the earth and soil, reducing the strength of the rock and soil and simultaneously generating dynamic and hydrostatic pressures and pore water pressures, exerting floating force on the rock and soil and increasing the weight of the rock and soil [25]. Thus, the distance to river has the second sensitivity of all factors. In the study area, most of the collapse of the loess hillside slope, or the initiation of loess landslides, after having been excavated during the construction of an expressway, was induced by excavation and rainfall [48]. The sensitivity of distance to road on the scale of the landslide is of third importance. In the construction of rural road projects, rough construction, such as cut-off or slope cutting and filling, is blindly carried out without the stability analysis of the slopes on both sides of the valley, which makes the slope steeper, loses support, forms a steep slope, and causes hidden dangers for the occurrence of landslide disasters [49]. Precipitation is the main recharge source for soil moisture in the hill and gully region of the Loess Plateau [50]. According to the survey, 40% of the catastrophic landslides are trigged by rainfall on the Loess Plateau of China [26]. It is not easy to infiltrate after precipitation, however, once the effective strength of the loess is reduced by rainfall, the landslide is triggered. There is a close correspondence between landslide occurrence and the pattern of precipitation, and 84.6% of landslides have happened in the monsoon in the Baota district since 1985 [51]. Therefore, rainfall is the fourth important factor in the study area. Elevation data reflected the location of the landslide, which means the volume of landslides has higher relative sensitivity on the variable of elevation in the study area. The slope is an important performance of the slope geometry, which determines the state and distribution of stresses in the slope mass and controls the stability and mode of the instability [51]. However, the slope has relatively little impact on the occurrence and scale of landslides in the study area. The increased vegetation coverage has a certain inhibitory effect on the amount of landslides, but considering the comprehensive effect of other factors, the influence of plants is not obvious. In addition, landslides with different scales occurred in different lithology. Thus, the influence of lithology on landslides is also relatively rare in the study area. The impact of environmental and human factors on the surface, such as construction activities, road cutting, and natural resources exploitation, may lead to landslides [34]. However, during the 9th Five-Year Plan period (1996–2000), Yangou watershed was selected as a demonstration region for integrated research for the development of ecological agriculture in the Loess Plateau, and the land use structure changed rapidly [41]. Nowadays, the land cover tends to be stable, so the absolute sensitivity of landslides to land use is lowest.

### 4.4. Results of the Landslide Susceptibility Mapping

Figures 4 and 5 showed the different space layers of conditioning factors and the landslide susceptibility map constructed by the VR model and RIRA method, respectively. In Figure 4, higher susceptibility values were in the layers with the convex slope and concave slope, as well as the distance of less than 50 m to the river, the distance of less than 50 m to the road, the slope of 25–35°, and the slope higher than 35°; the values were all more than 10. Among the nine related factors examined in the study, elevation, distance to river, lithology, and rainfall are hard to change, and the other factors, slope, profile curve, distance to road, land-use, and NDVI, can be modified to reduce the landslide

susceptibility. Slope gradients can be reduced by converting slope land into terraces [21]. In terms of profile curve and distance to road, human activity and road construction should be carried out in a planned way. Especially, a preliminary reconnaissance survey should be implemented on the topography and geology of the area. In terms of land use, many landslides were observed on the bareland. Thus, the reduction of ground exposure may reduce landslide susceptibility. For the NDVI, it can be seen that soil bioengineering is not omnipotent and did not apply to all situations, e.g., deep-seated, rock, or extremely steep slopes [52]. Therefore, the comprehensive effect of the conditioning factors should be considered in the process of prevention and control of landslides. In the study area, reconstruction of the eco-environment had already been conducted since 1997. In the past decade, grasses or shrubs are the main types of plantings on the top of hills or upper part of hill slopes, while trees were mainly selected in the hilly slope with a moderate gradient, and most of the slope of cropland was converted to woodland or grassland [53]. Nevertheless, a quantitative assessment of the effectiveness of those comprehensive measures in inhabiting landslides is still lacking.

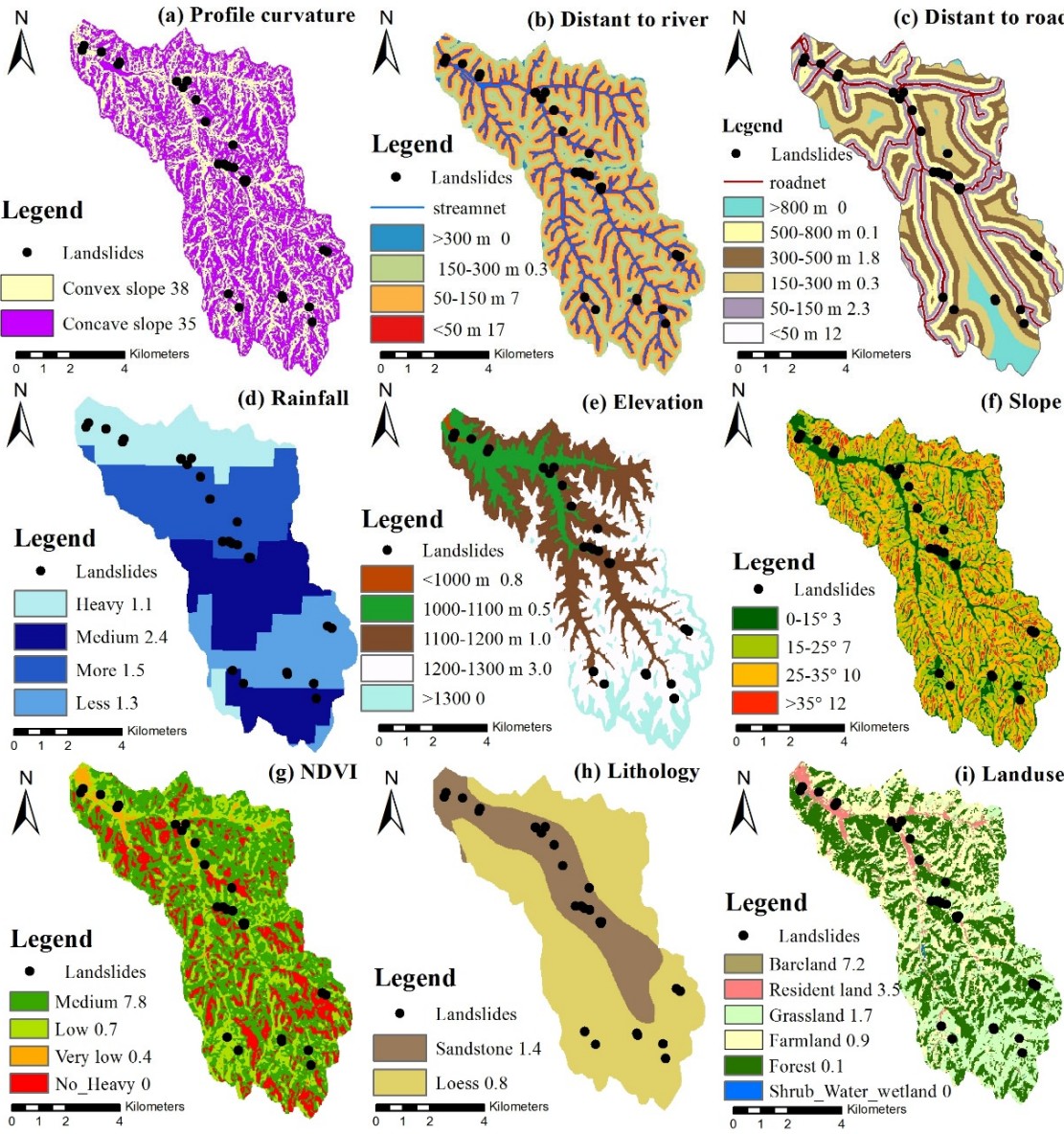

**Figure 4.** Susceptibility value of spatial layers influencing landslide occurrence in the study area: (**a**) profile curvature, (**b**) distance to river, (**c**) distance to road, (**d**) rainfall, (**e**) elevation, (**f**) slope, (**g**) NDVI, (**h**) lithology, and (**i**) land use.

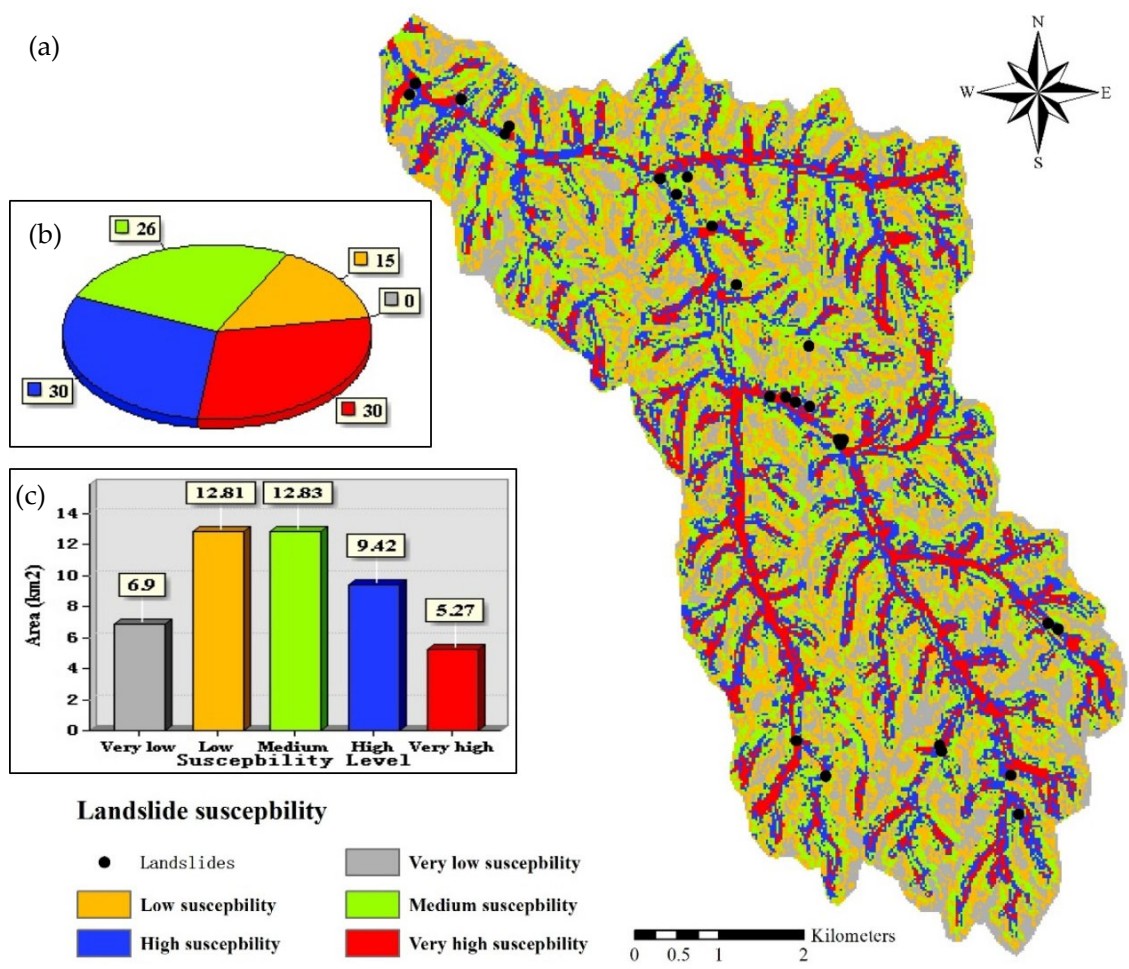

**Figure 5.** Landslide susceptibility map by the VR model and RIRA method: (**a**) landslide susceptibility classes in the Yangou watershed, (**b**) distribution of susceptibility over the landslide at susceptibility map (%), and (**c**) histogram of landslide susceptibility index value.

As shown in Figure 5c, a very low susceptibility zone occupies 6.90 km$^2$. Similarly, low, medium, high, and very high susceptibility zones occupy 12.81, 12.83, 9.42, and 5.87 km$^2$, respectively. As depicted in Figure 5a, the susceptibility levels decreased with distance from rivers and road. Very high susceptibility was in the region with a larger relief and along the river and road, which was also in line with the site investigation. In the whole province, landslides were mostly distributed on both sides of the roads and rivers. For example, collapses often take place on both sides of the Yan and the Fenchuan Rivers, especially along the roads and railways, [51]. This is due to the fact that Yan'an City and urban population centers are located in major river valleys and that intensive human engineering activities are often responsible for many landslides and related serious damages [51]. Therefore, in the road construction and other engineering works on the Loess Plateau, the engineers and researchers should be committed to the prevention and control of loess landslides [48], and, also, the road and the river located in the gullies. As a result, the down-cutting of channels during rainstorms, as well as the high content of soil moisture adjacent to a channel, make it more prone to landslides [21]. Moreover, in the regional susceptibility assessment of the Baota district, the specific area was extended along the Nanchuan River, and its tributaries to Shitougou village got a high susceptibility [28]. Figure 5b revealed that no landslide is located within the very low susceptibility areas; 15% of the landslide are situated within the low susceptibility areas; 26% within the moderate suitability areas; 30% within the high susceptibility areas; 30% within the very high susceptibility areas. The result of the landslide assessment highlighted that the majority of the occurred landslides

(60%), are situated within the zones of high and very high susceptibility, while the part of landslide volume was accounting for 87% of the total landslides. Landslide susceptibility is predicting the probability of landslides' occurrence regardless of their magnitude which is, however, a critical piece of information in mass movement erosion control. However, the previous study revealed small and medium mass movements often have much higher transport rate of debris than large ones [21]. Both the volume and frequency of landslides were considered in our study; the landslide mitigation and control areas can be further prioritized in the watershed from the map. The high and very high susceptibility area must receive higher priority in the watershed for large landslides, while the low susceptibility risk area can be excluded in the observation. Moreover, the area of low and medium susceptibility can be paid more attention to, as well as the soil erosion caused by landslides. The results of a zoning plan can also reduce the workload of field investigations and move the focus of the actual work in watershed management.

## 5. Conclusions

To understand the role of landslide disasters in small watersheds more comprehensively, a landslide susceptibility assessment that considers the amount of landslides is necessary. The applied analysis of the VR model, RIRA method, and ArcGIS within the framework of the present study achieved more comprehension for disaster prevention and land planning with respect to the location and volume of the past landslides. The relationship among landslide occurrence, landslide volume, and nine landslide conditioning factors such as slope, elevation, NDVI, land use, lithology, distance to road, distance to river, profile curvature, and rainfall are evaluated using the above models. Our analysis demonstrated that the sub-classes in the same conditioning factors had different VR values, as indicated that sub-regions with homogenous properties contributed differently to the landslide scales. In addition, the results showed that the various parameters of landslides had different sensitive values to the landslide volume and occurrence. Among those factors, profile curvature and distance to river were the two most highly sensitive, followed by distance to road, rainfall, elevation, slope, and NDVI. The lithology and land-use were the two lowest sensitive factors. The sensitive values are 0.3616, 0.2567, 0.0908, 0.0628, 0.0523, 0.0345, 0.0203, 0.0099, and 0.0097, respectively. Also, the landslide susceptibility map considering landslide volume classified the study area into five zones, with susceptibility degrees of very high, high, medium, low, and very low, which showed that about 60% of the landslides occurred in a high and very high susceptibility area, accounting for 87% of the total volume of landslides. Landslides such as slides and falls are important contents of geological disaster prevention and control, and this phenomenon is also the main form of soil erosion. Small landslide erosion is more likely to cause sediment yield. Thus, the spatial distribution of the existing landslide susceptibility can be analyzed in order to identify the components that are located in large landslide scale areas and small landslide scale areas, so as to set the priorities to prevent and control landslide disasters and soil erosion. Engineers, decision-makers, and environmental managers may implement the analysis that we implemented in the present study during new or existing planning projects and produce maps that will make possible the adoption of policies and strategies aimed at multi-hazard mitigation.

**Author Contributions:** Writing—original draft preparation, H.G.; writing—review and editing, X.Z. All authors have read and agreed to the published version of the manuscript.

**Funding:** This research was funded and supported by the National Natural Science Foundation of China (Grant No: 52009103, 42177346) and the Innovation Capability Support Program of Shaanxi (2019TD-040).

**Institutional Review Board Statement:** Not applicable.

**Informed Consent Statement:** Not applicable.

**Data Availability Statement:** Not applicable.

**Acknowledgments:** We are grateful to the anonymous reviewers for their valuable suggestions in improving the manuscript.

**Conflicts of Interest:** The authors declare no conflict of interest.

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
