# Peer review of "Landslide Susceptibility Assessment Considering Landslide Volume: A Case Study of Yangou Watershed on the Loess Plateau (China)"

_applsci, doi:10.3390/app12094381_

Round 1

Reviewer 1 Report

-my major point is insufficient details in conclusion.

-insufficient highlight of lack of the area

-method needs to be explained further and most importantly supported by published literature

Please see file attached.

Reviewer 2 Report

The authors have to improve the manuscript by considering the following points:

  1. English should be improved
  2. Proper classification of landslides should be adopted
  3. Landslide volume calculation should be given in methods
  4. Geology of the study area has to be given either in the introduction or study area section
  5. The discussion part is too superficial. Rewrite the discussion part.
  6. I have attached the annotated version of the MS
